# Prevalence of *mecA* and *Panton-Valentine Leukocidin* Genes in *Staphylococcus aureus* Clinical Isolates from Gaza Strip Hospitals

**DOI:** 10.3390/microorganisms11051155

**Published:** 2023-04-28

**Authors:** Nabil Abdullah El Aila, Nahed Ali Al Laham, Thierry Naas

**Affiliations:** 1Department of Medical Laboratory Sciences, Faculty of Medical Sciences, Al-Aqsa University Gaza, Gaza P.O. Box 405, Palestine; 2Department of Laboratory Medicine, Faculty of Medical Sciences, Al-Azhar University-Gaza, Gaza P.O. Box 1277, Palestine; 3Bacteriology-Hygiene Unit, Hôpital Bicêtre, AP-HP Paris-Saclay, 94270 Le Kremlin-Bicêtre, France; 4Faculty of Medicine, Team ReSIST, UMR1184, INSERM, CEA, Université Paris-Saclay, 94270 Le Kremlin-Bicêtre, France; 5French National Reference Center for Antimicrobial Resistances, Hôpital Bicêtre, AP-HP Paris-Saclay, 94270 Le Kremlin-Bicêtre, France

**Keywords:** *Staphylococcus aureus*, MRSA, MSSA, PVL, PCR, antimicrobial susceptibility testing, Gaza Strip, Palestine

## Abstract

Methicillin-resistant *Staphylococcus aureus* (MRSA) are spreading worldwide in hospital and community settings, thus posing a serious public health problem. Panton-Valentine Leukocidin (PVL), an important virulence factor of *S. aureus*, is a marker of community-acquired MRSA. Here we determined the prevalence of *pvl* genes among *S. aureus* isolates from different hospitals in the Gaza Strip, Palestine. A total of 285 *S. aureus* isolates were collected from five different hospitals in the Gaza Strip. All isolates were characterized for their susceptibility patterns to available antimicrobial agents and by using multiplex PCR for the detection of *mecA* and *pvl* genes. The overall prevalence of MRSA in Gaza hospitals was 70.2% (range: 76.3% to 65.5%) and that of *pvl* among *S. aureus* isolates was 29.8% (range: 32.9% to 26.2%). The *pvl* gene was equally prevalent among MRSA isolates (30.5%) and MSSA isolates (28.2%). The most effective antibiotics were rifampicin, vancomycin, and clindamycin, with susceptibility rates of 91.2%, 88.7%, and 84.6%, respectively. The highest percentage of strains were observed to be resistant to penicillin and amoxicillin with clavulanic acid—96.1% and 73.6%, respectively. Our results showed a high prevalence of MRSA and *pvl*-positive isolates in Gaza Strip hospitals, which likely reflects the situation in the community. It is mandatory to implement systematic surveillance of both hospital and community isolates, together with interventions (such as increased hand hygiene, use of hydroalcoholic solutions, and isolation of carriers) to limit their spread.

## 1. Introduction

*Staphylococcus aureus* (SA) is a major human pathogen that causes infections ranging from minor skin and soft tissue infections to life-threatening sepsis. It has been linked to both community-acquired (CA) and healthcare-associated (HA) infections, both of which cause significant morbidity and mortality [1,2,3,4]. Methicillin-Resistant SA (MRSA) isolates were initially identified in 1961 and have since become major nosocomial pathogens worldwide [5]. Methicillin resistance developed because of the acquisition of a mobile genetic element known as staphylococcal cassette chromosome *mec* (*SCCmec*), which contains the *mecA* resistance gene that encodes a penicillin-binding protein variant (PBP2a) with a lower affinity for β-lactam antibiotics [6]. A new variant of *mecA*, known as *mecC* (initially referred to as mecALGA251), has been discovered in MRSA samples collected from humans and various animal species, including livestock, small mammals, and birds in Europe. This type of *mec* bears 70% similarity in terms of nucleotide identity with *mecA* (Harrison et al., 2014).

The ability of MRSA to cause various infections is primarily due to the presence of toxins and various extracellular and surface virulence factors with adhesive properties, such as Panton-Valentine Leukocidin (PVL) [7]. By creating pores in the mitochondrial membrane, PVL destroys leukocytes and alters the immune system, ultimately leading to mitochondrial cell membrane lysis and death [8,9]. The PVL toxin is responsible for skin and soft tissue infections, necrotizing pneumonia, diffuse cellulitis, and osteomyelitis [10]. Patients with PVL-positive MRSA infection had a higher mortality rate [11]. Although the prevalence of PVL-producing *S. aureus* isolates varies across the globe, increasing reports signal that the increase in PVL-positive MRSA constitutes a significant public health challenge [12].

Epidemiological and clinical data show that CA-MRSA isolates with high virulence potential express PVL [13,14], which has also been shown to be an important contributing factor in CA-MSSA infections, especially in severe skin and soft tissue infections (SSTI) and, most remarkably, necrotizing pneumonia [15,16,17]. The *pvl* genes were discovered in cutaneous and invasive infections carried by HA-MRSA strains [18]. According to the published data in Palestine, the prevalence of resistance to methicillin among *S. aureus* strains isolated from hospitalized patients is steadily increasing [19,20,21,22,23]. However, only a few studies in Palestine, and almost none in the Gaza Strip, addressed the prevalence and clonal spread of PVL-producing *S. aureus* isolated from patients in Palestine. 

Our study aims to provide insights into the molecular characteristics of PVL-producing *S. aureus* strains and antibiotic resistance phenotypes considering the increasing prevalence of Multidrug-Resistant (MDR)-PVL-positive MRSA and its spread in Gaza Strip hospitals. 

## 2. Materials and Methods

### 2.1. Study Design 

The Gaza Strip (31°250 N, 34°200 E) is a narrow territory along the eastern Mediterranean coast (41 km long and 6–12 km wide), with tightly controlled borders abutting Israel and Egypt’s Sinai Peninsula. It is regarded as one of the world’s most densely populated areas, with a population of approximately 2.1 million inhabitants (37.5% of the total estimated Palestinian population) and a population density (4073/km^2^) nearly ten times greater than that of the West Bank (433/km^2^) [24].

The study was set out as a cross-sectional study in five different hospitals in the Gaza Strip. 

Two-hundred and eighty-five *S. aureus* isolates were collected from different clinical samples including pus, blood, sputum, urine, nasal, and ear swab samples between July 2019–December 2019. The study was approved by the department of human resources and development in the Ministry of Health—Gaza, Palestine under study approval number PHRC/HC/227/17.

### 2.2. Sample Collection and Phenotypic Detection of S. aureus

In each hospital involved in this study, *S. aureus* isolates were collected and identified using morphological (Gram stain), biochemical (Coagulase, Mannitol salt agar, and DNase) methods. To confirm the fermentation of mannitol, the growth of yellow colonies on Mannitol salt agar (HiMedia, Mumbai, India) surrounded by yellow zones after 24 h of incubation at 37 °C indicated a positive result [25]. For tube coagulase tests, colonies of test isolates were re-suspended in 2 mL of citrated human plasma in sterile glass test tubes. The tubes were incubated at 35 °C for 4 h and observed for clot formation. The DNase test was performed by inoculating Staph isolates on deoxyribonuclease (DNase) agar (HiMedia, Mumbai, India) and incubated for 24 h at 37 °C. An excess of 1 N HCl (about 15 mL) was then added. A vacuum pipette was used to remove extra acid and clear areas around the bacterial colonies revealed DNase-positive colonies [25]. The isolates were kept in trypticase soy agar until further investigation. Confirmation of identification and antibiotic susceptibility testing of the *S. aureus* strains was carried out in the microbiology laboratory of Al Aqsa University in the Gaza Strip, Palestine.

### 2.3. DNA Extraction and Molecular Detection of mecA Gene and lukS/F-PV Gene by PCR 

DNA was extracted from cultured isolates using alkaline lysis, as previously described [26,27]. In brief, one bacterial colony was suspended in 20 µL of lysis buffer (0.25% sodium dodecyl sulfate, 0.05 N NaOH) and heated for 15 min at 95 °C. A total of 180 µL of distilled water was used to dilute the cell lysate. The cell debris was centrifuged at 16,000× *g* for 5 min, and the supernatants were used for PCR or stored at −20 °C until further use. Extracted DNA from the different isolates was used in a multiplex PCR assay that detects three genes: the 16SrRNA gene used as an internal control, the *mecA* gene for MRSA detection, and the *lukS/F-PV* gene for PVL detection [27]. Two control strains, the *S. aureus* ATCC 25923 (carrying the *pvl* gene) and *S. aureus* ATCC 33592 (carrying the *mecA* gene), were added in each run.

Thermal cycling conditions were set at 94 °C for 10 min followed by 10 cycles of 94 °C for 45 s, 55 °C for 45 s, and 72 °C for 75 s and 25 cycles of 94 °C for 45 s, 50 °C for 45 s, and 72 °C for 75 s. PCR products were analyzed on 1.5% agarose gel containing 2 µL ethidium bromide (0.5 µg/mL). The primers are listed in Table 1 [27].

### 2.4. Antimicrobial Susceptibility Testing

Antibiotic susceptibility testing was performed following the Kirby Bauer disc diffusion method as recommended by CLSI guidelines [28] using Mueller–Hinton agar. Before inoculation, the swab stick was dipped into bacterial suspension with visually equivalent turbidity to 0.5 McFarland standards. Antibiotic discs (Himedia, India) tested included penicillin G (10 μg), vancomycin (30 μg), erythromycin (15 μg), chloramphenicol (30 μg), co-trimoxazole (25 μg), rifampicin (30 μg), clindamycin (2 μg), amoxicillin-clavulanic acid (20/10 μg), tetracycline (30 μg), ciprofloxacin (5 μg), and cefoxitin (30 μg). The zones of inhibition for each antimicrobial agent were measured and interpreted as resistant, intermediate, or susceptible according to local interpretation rules used in the hospitals of the Gaza strip, which are derived from the CLSI guidelines when available from the technical data sheets of Himedia, or correspond to the former inhibition zone diameter but are still in use in Gaza hospitals to infer antibiotic susceptibility. Thus, co-trimoxazole, vancomycin, and amoxicillin-clavulanic acid were not interpreted using CLSI nor EUCAST guidelines, but using the following zone diameters: R < 10 mm and S ≥ 16 mm for co-trimoxazole; R < 19 mm and S ≥ 20 mm for amoxicillin-clavulanic acid; and R < 17 mm and S ≥ 17 mm for vancomycin. Methicillin resistance was inferred using the zone diameter around cefoxitin following CLSI guidelines [28].

### 2.5. Statistical Analysis 

The results were tabulated and analyzed using the Statistical Package for the Social Sciences (SPSS, version 20). Frequencies, cross-tabulation, and appropriate statistical tests such as the Chi-square test and Fisher’s exact test were performed. A *p*-value of less than 0.05 was considered significant.

## 3. Results

### 3.1. Molecular Analysis

A total of 285 *S. aureus* isolates were collected from five different hospitals in the Gaza Strip including Al Naser, Al Shifa, Al Aqsa, Europeans, and Nasser hospitals. The number of *S. aureus* isolates collected from the preceding hospitals was 85, 72, 61, 41, and 26, respectively (Table 2).

The majority of the *S. aureus* isolates were collected from pus (208/285 [72.9%]), followed by blood (31/285 [10.8%]), sputum (17/285 [5.9%]), urine (14/285 [4.9%]), and other sites of infection such as eye secretion, ear secretion, nasal, skin, umbilical cord, and vaginal discharge (15/285 [5.2%]) (Table 3).

PCR analysis of the 285 clinical *S. aureus* isolates revealed 200 *mecA*-positive isolates (70.2% MRSA) while 85 were *mecA*-negative (29.8% MSSA). The prevalence of MRSA in Al Shifa, Al Naser, Al Aqsa, European, and Nasser hospitals was 76.4%, 70.5%, 65.8%, 69.2%, and 65.6%, respectively. *Pvl* toxin gene was detected in 85 (29.8%) of all tested isolates. The prevalence of PVL in Al Naser, Al Aqsa, Al Shifa, European, and Nasser hospitals was 32.9%, 31.7%, 29.1%, 26.9, and 26.2%, respectively. The distribution of PVL-positive *S. aureus* isolates according to the clinical specimens was as follows: pus 70 (33.6%), sputum 6 (35.2%), blood 6 (19.3%), other samples 2 (13.3%), and urine 1 (7.1%), respectively (Table 3). The prevalence of the *pvl* gene among MRSA and MSSA isolates was 30.5% and 28.2%, respectively. 

### 3.2. Antimicrobial Susceptibility Testing

Disk diffusion antimicrobial susceptibility testing revealed that the least effective antibiotics were penicillin G and amoxicillin-clavulanic acid with resistance rates of 96.1% and 73.6%, respectively (Table 4). The most effective antibiotics were rifampicin, vancomycin, and clindamycin, with susceptibility rates of 91.2%, 88.7%, and 84.6%, respectively. Nearly half of the isolates were resistant to erythromycin (47.7%).

When considering PVL+ isolates, PVL+ MRSA isolates were highly resistant to penicillin G and amoxicillin-clavulanic acid (95% and 83.5%, respectively) but have a low prevalence of resistance to rifampicin, clindamycin, chloramphenicol, and vancomycin (Table 5). PVL+ MSSA isolates have a low prevalence of resistance to amoxicillin-clavulanic acid, vancomycin, co-trimoxazole, rifampicin, and ciprofloxacin in comparison with PVL+ MRSA (Table 5). 

The prevalence of MDR, e.g., resistance to at least three families of antibiotics, among *S. aureus* isolates was 66% (188/285). The prevalence of MDR among MRSA and MSSA isolates was 75.5% (151/200) and 43% (37/85), respectively.

## 4. Discussion

Panton-Valentine Leukocidin (PVL) is a bicomponent pore-forming cytolytic toxin encoded by the *lukF/PV* and *lukS/PV* genes. It was thought as one of the major virulence factors contributing to the morbidity and mortality attributed to *S. aureus* [14,17,27,29] causing skin and soft tissue infections, such as cutaneous abscesses and severe necrotizing pneumonia. However, the role of PVL toxin in determining the severity of MRSA infections has been and still is controversial. Newer pieces of evidence showed that the presence of *pvl* genes is neither the primary cause nor the determinant of MRSA infection severity [30,31,32]. Other studies have found that PVL-positive isolates have a high proclivity for invasive infections, but the presence of *pvl* genes is not the main determinant of MRSA infection severity nor invasivity [33,34]. As a result, it is important to investigate the prevalence of the PVL marker among MRSA isolates, which are a major health concern due to their multidrug resistance [35,36].

In this study, MRSA accounted for 70.2% (*n* = 200) of the isolates tested by PCR for the *mecA* gene and 69.1% (*n* = 197) according to cefoxitin sensitivity testing. This discrepancy may be due to the fact that molecular testing is more sensitive and specific for the diagnosis of MRSA. So, we suggest that molecular approaches should be favored. Moreover, one limitation of this study is that only *mecA* genes were thought of and that even though rare, future evaluations need to include the *mecC* gene. The prevalence in our study was higher than that reported by previous studies conducted in Gaza and the West Bank (56.3%, 29%, and 45%, respectively) except for one that reported a slightly higher prevalence (82.3%), which could be explained by the phenotypic detection method used (oxacillin discs) which is known to be less specific than molecular testing [20,21,22,23,37]. Our results are comparable to those reported in Jordan, Egypt, and Cyprus, where more than 50% of the invasive *S. aureus* isolates were methicillin-resistant [38]. Tabaja et al. reported that the available data suggest a high burden of MRSA in Arab countries of the Middle East and North Africa (MENA) [39]. The prevalence of MRSA (76.3%) in Al Shifa hospital, which is considered the major referral hospital in the Gaza Strip, was higher than that reported from the same hospital by Al Laham et al. (56.3%) in 2015 [20]. Our results were consistent with those reported in Al-Karak hospital and Prince Ali hospital in Jordan (77.5%) [40]. In a recent study in Egypt, a high prevalence of MRSA isolates was identified (138/170, 81.2%) with 79% of isolates (109/138, 79%) being MDR-MRSA [41]. 

The frequency of PVL-harboring isolates (PVL+) in our study was 29.8%, which is lower than the 40% reported by Al Laham et al. [20]. The highest rate of PVL among hospitals in the Gaza Strip was found in Al Naser and Al Aqsa hospitals (32.9% and 31.7% respectively). In comparison with previous studies that were conducted in Palestine, our results are consistent with those of two hospitals from the West Bank and East Jerusalem [23] (29.5%) but higher than those of Rafidia and Thabet hospitals (West Bank) [37] and from healthy children throughout the Gaza Strip [21] (14.3% and 8.5%, respectively). These differences in PVL prevalence could be a reflection of the different dominant MRSA clones circulating in these regions that may or may not harbor *pvl* genetic determinants.

Many studies from the Arab countries of the Middle East investigated the molecular epidemiology of clinical *S. aureus* including detection of the *pvl* gene. In Saudi Arabia and Alexandria, Egypt, *pvl* prevalence (48.7% and 40%, respectively) was higher than our findings (29.8%) [42,43]. Meanwhile, a recent study from Kuwait hospitals showed only 21.6% *pvl*-positive *S. aureus* isolates [44] and, surprisingly, a study conducted in Oman has revealed the dominance of a *pvl*-negative clone (ST6-IV/t304) [45].

Finally, our results are in line with a study in Greece that reported a frequency of 27% of PVL genes among MRSA [46], while Omuse et al. from Kenya [47] reported higher rates (42.3%) and Shariati et al. [48] from Iran reported only 10.7%. Subarna Roy et al. from India, have reported an overall 62.85% of PVL prevalence among MRSA and MSSA (MRSA: 85.1% and MSSA: 48.8%) [49]. Other parts of the world reported a lower prevalence of PVL (5% in France, 4.9% in the UK, 8.1% in Saudi Arabia, and 14.3% in Bangladesh [17,50,51,52]), indicating that the prevalence of PVL varies greatly between geographical locations and populations.

In our study, PVL was detected in 30.5% (61/200) of MRSA isolates and in 28.2% (24/85) of MSSA isolates. Our results showed no significant difference between MRSA and MSSA populations in terms of PVL carriage. In Nepal, Algeria, Bangladesh, Greece, and Romania, the prevalence rates of the *pvl* gene among MSSA were 26%, 16.4%, 27.3%, 12%, and 14%, respectively [46,50,53,54,55]. In Iran, Shariati et al. [48] reported PVL carriage in 18.8% of MRSA and only 3% of MSSA, which is in line with other studies that reported higher prevalence in MRSA isolates than MSSA [1,56,57,58]. However, this is not a general rule, as Alli et al. in Nigeria reported a higher prevalence of the *pvl* gene in MSSA (53.3%) than in MRSA isolates (9.1%) [59]. This is also consistent with Azar et al., who showed that the prevalence of the *pvl* gene in MRSA strains was 7.23%, while that in MSSA was 33.3% [60]. 

PVL is associated with increased virulence of certain *S. aureus* strains and is the cause of necrotic lesions affecting the skin and mucosa that are extremely difficult to treat, such as necrotic hemorrhagic pneumonia. As a result, early detection of the *pvl* gene in *S. aureus* may be critical for treating cases and assessing treatment outcomes.

MRSA was found in 72.2% (150/208 isolates) of *S. aureus* from pus. The majority of our PVL-positive isolates were isolated from pus and blood samples (72% and 64%, respectively). 

*S. aureus* has a positive relationship with skin and soft tissue infections (SSTIs); in fact, most investigations have found that the frequency of the PVL gene is higher in pus specimens collected from SSTIs compared to blood, urine, or sputum samples [51]. In this study, the prevalence of the *pvl* gene among sputum, blood, and urine isolates was 60%, 30%, and 11%, respectively.

Kamarehei, et al. [61] reported in Iran that the majority of PVL-carriage isolates (30.4%) were found in wound specimens, and 27.6% of urine specimens were PVL positive, which could play a role in virulence. Singh-Moodley et al. [62] characterized MRSA isolates from blood cultures in South Africa from 2013 to 2016, reporting a PVL positivity rate of 25%, which was lower than the rate in our study.

Our results showed that the least effective antibiotics were penicillin G and amoxicillin-clavulanic acid. Their resistance rate was 96.1% and 73.6%. Demir et al. from Turkey [63] reported a higher rate of sensitivity for amoxicillin-clavulanic acid (53.6%) than our study (26.3%). The resistance rate of PVL-producing MRSA and MSSA strains against penicillin G was 95%. These results are comparable to those of Al Laham et al. [20] in Palestine and Bazzi et al. in Dhahran, Saudi Arabia, which reported 98.9% and 88.75%, resistance, respectively [64]. 

Surprisingly, only 84.3% of MRSA isolates were susceptible to vancomycin. This result is consistent with that of Islam et al. who reported a susceptibility rate of 86%, but different from other studies in the Gaza Strip and West Bank that reported 100% susceptibility to vancomycin [20,21,23]. Moreover, this discrepancy may be due to visual MacFarland determination because we did not use a nephelometer to determine the 0.5 MacFarland, or it was a reading of one or two millimeters above the CLSI breakpoints. 

This implies that vancomycin remains an effective treatment option and is considered a last-resort antibiotic for severe MRSA and other resistant Gram-positive infections [65,66,67,68,69]. The sensitivity rate of MRSA isolates against clindamycin was 88%. Clindamycin has been recommended for the treatment of cutaneous infections related to *S. aureus* [63].

## 5. Conclusions

In comparison to previously published results from the Gaza Strip, our study highlights the fact that the rate of multidrug-resistant MRSA among the population is increasing in Palestine as 70.2% of our *S. aureus* (which were MRSA) isolates were MDR. 

Vancomycin, erythromycin, and clindamycin are common examples of antibiotics used for the treatment of infections caused by Gram-positive bacteria, and the susceptibility of the *S. aureus* clinical isolates in our study to these antimicrobial agents demonstrated that these antibiotics are still effective for the treatment of infections caused by these pathogens.

We reported that 29.8% of MRSA isolates from hospitalized patients from five different hospitals in the Gaza Strip were PVL positive. As PVL is a pathogenicity marker, these PVL-positive isolates are likely more virulent and must be closely monitored and treated. 

Thus, given the high prevalence of MRSA, it is critical to screen clinical samples for *mecA*, but also for *pvl* genes to properly guide therapy. In addition, continued surveillance and characterization of MRSA isolates in Gaza Strip hospitals are critical for preventing the spread of virulent nosocomial infections and implementing enhanced infection control strategies.

## Figures and Tables

**Table 1 microorganisms-11-01155-t001:** PCR primers used in this study.

Primers	Gene Targeted	Sequence	Product Size (bp)
Staph756F	16SRNA	5-AACTCTGTTATTAGGGAAGAACA-3	
Staph750R	16SRNA	5-CCACCTTCCTCCGGTTTGTCACC-3	756
Leuk-PV-1	*lukS/F*	5-ATCATTAGGTAAAATGTCTGGACATGATCCA-3	
Leuk-PV-2	*lukS/F*	5-GCATCAAGTGTATTGGATAGCAAAA GC-3	433
Mec-A1	*mecA*	5-GTAGAAATGACTGAACGTCCGATAA-3	
Mec-A2	*mecA*	5-CCAATTCCACATTGT TTCGGTCTAA-3	310

**Table 2 microorganisms-11-01155-t002:** Prevalence of *mecA* and *pvl* genes according to the hospitals in the Gaza Strip.

	MRSA	MSSA	PVL	MRSA PVL+	MRSA PVL−	MSSA PVL+	MSSA PVL−
Al Shifa (72)	55 (76.3%)	17 (23.6%)	21 (29.1%)	15 (20.8%)	40 (55.5%)	6 (8.3%)	11 (15.2%)
Al Nasser (85)	60 (70.5%)	25 (29.4%)	28 (32.9%)	21 (24.7%)	39 (45.8%)	7 (8.2%)	18 (21.1%)
Al Aqsa (41)	27 (65.8%)	14 (34.1%)	13 (31.7%)	8 (19.5%)	19 (46.3%)	5 (12.1%)	9 (21.9%)
European (26)	18 (69.2%)	8 (30.8%)	7 (26.9%)	6 (23%)	12 (46.1%)	1 (3.8%)	7 (26.9%)
Naser (61)	40 (65.6%)	21 (52.5%)	16 (26.2%)	11 (18%)	29 (47.5%)	5 (8.1%)	16 (26.2%)
Total (285)	200 (70.1%)	85 (29.8%)	85 (29.8%)	61 (21.4%)	139 (48.7%)	24 (8.4%)	61 (21.4%)

**Table 3 microorganisms-11-01155-t003:** Prevalence of *mecA* and *pvl* genes according to the sample type.

	MRSA	MSSA	PVL	MRSA/PVL+	MRSA/PVL−	MSSA/PVL+	MSSA/PVL−	Total/Sample Type
Blood	20	11	6	4	16	2	9	31
Sputum	10	7	6	4	6	2	5	17
Pus	150	58	70	51	99	19	39	208
Urine	9	5	1	1	8	0	5	14
Others	11	4	2	1	10	1	3	15
Total	200	85	85	61	139	24	61	285

**Table 4 microorganisms-11-01155-t004:** Antibiotic Susceptibility profile of *S. aureus*.

	R ^$^		S ^$^		I ^$^	
	No.	%	No.	%	No.	%
Penicillin G	274	96.1	11	3.9	0	0
Amoxicillin/ Clavulanic Acid *	210	73.6	75	26.3	0	0
Cefoxitin	198	69.5	87	30.5	0	0
Erythromycin	136	47.7	132	46.3	17	6
Co-trimoxazole *	92	32.3	185	64.9	8	2.8
Tetracycline	79	27.2	206	72.3	0	0
Ciprofloxacin	75	26.2	193	67.7	17	6
Chloramphenicol	52	18.2	229	80.4	4	1.4
Vancomycin *	32	11.2	253	88.7	0	0
Clindamycin	31	10.9	241	84.6	13	4.5
Rifampicin	23	8.1	260	91.2	2	0.7

^$^ Legend: R: Resistant. S: sensitive, I: Intermediate. * For these antibiotics, antibiotic susceptibility was not interpreted following CLSI nor EUCAST guidelines but using former CLSI inhibition diameters that are still in use in Gaza hospitals. Thus, the following zone diameters were used: R < 10 mm and S ≥ 16 mm for co-trimoxazole; R < 19 mm and S ≥ 20 mm for amoxicillin-clavulanic acid; and R < 17 mm and S ≥ 17 mm for vancomycin.

**Table 5 microorganisms-11-01155-t005:** Antibiotic resistance profile of *S. aureus* according to *mecA* and *pvl* positivity.

	MRSA (*n* = 200)	MSSA (*n* = 85)	MRSA/PVL+ (*n* = 61)	MRSA/PVL− (*n* = 139)	MSSA/ PVL+ (*n* = 24)	MSSA/ PVL− (*n* = 61)
Penicillin G	197 (98.5%)	81 (95.2%)	58 (95%)	139 (100%)	23 (95.5%)	58 (95%)
Amoxicillin/ Clavulanic Acid *	160 (80%)	53 (62%)	51 (83.6%)	108 (77.6%)	13 (54.5%)	40 (65.5%)
Cefoxitin	197 (98.5%)	80 (32.9%)	59 (96.7%)	139 (100%)	23 (95.5%)	57 (93.4%)
Erythromycin	95 (47.5%)	39 (45.8%)	29 (47.5%)	66 (47.5%)	10 (41.6%)	29 (47.5%)
Co-trimoxazole *	73 (36.5%)	17 (20%)	20 (32.7%)	53 (38.1%)	1 (4.1%)	16 (26.2%)
Tetracycline	48 (24%)	18 (21%)	19 (31.1%)	40 (28.7%)	6 (25%)	12 (18.7%)
Ciprofloxacin	60 (30%)	15 (17.6%)	17 (27.8%)	38 (27.3%)	1 (4.7%)	13 (21.3%)
Chloramphenicol	34 (17%)	18 (21.1%)	8 (13.1%)	26 (18.7%)	5 (20.8%)	13 (21.3%)
Rifampicin	21 (10.5%)	2 (2.3%)	5 (8.1%)	16 (11.5%)	1 (4.1%)	1 (1.6%)
Vancomycin *	31 (15.5%)	10 (5%)	10 (16.3%)	16 (11.5%)	2 (8.3%)	8 (13%)
Clindamycin	24 (12%)	6 (7%)	5 (8.1%)	19 (13.6%)	2 (8.3%)	4 (6.5%)

* For these antibiotics, antibiotic susceptibility was not interpreted following CLSI nor EUCAST guidelines but using former CLSI inhibition diameters that are still in use in Gaza hospitals. Thus, the following zone diameters were used: R < 10 mm and S ≥ 16 mm for co-trimoxazole; R < 19 mm and S ≥ 20 mm for amoxicillin-clavulanic acid; and R < 17 mm and S ≥ 17 mm for vancomycin.

## Data Availability

All data generated or analyzed during this study were included in this article.

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
