# Peer review of "Prevalence of *mecA* and *Panton-Valentine Leukocidin* Genes in *Staphylococcus aureus* Clinical Isolates from Gaza Strip Hospitals"

_microorganisms, 2023, doi:10.3390/microorganisms11051155_

Round 1

Reviewer 1 Report

The main focal point of this paper is about the association between PVL toxin and the severity of MRSA infections, even though this claim is highly controversial. While some studies have suggested that MRSA strains that produce PVL are more virulent and cause more severe infections, others have found no significant difference in the severity of infections caused by PVL-positive and PVL-negative strains. Furthermore, there is also evidence that PVL alone may not be the main factor associated with the severity of the infection and other virulence factors may play a more important role. Therefore, the relationship between PVL and the severity of MRSA infections is still a subject of ongoing research and debate and is not absolute. Therefore, the Title of this article should be redirected from PVL association and should focus more on the high prevalence of MRSA among S. aureus clinical isolates and the accompanying antibiotic resistance profiles identified. 

Major comments/requests/questions

·      The lack of molecular typing data about the isolates is a major limitation of this study. Therefore, the authors are requested to carry out sequence typing, spa typing and SCCmec typing for the isolates. This will provide a snapshot of the S. aureus strains circulating in Gaza and whether these strains are evolving. Also, the typing will provide some information about the origin of these isolates and whether they are likely to be community or hospital acquired. 

·      According to the 2021 CLSI guideline, vancomycin antibiotic has no zone diameter breakpoints for differentiating Resistant isolates from susceptible ones. How were the results interpreted?

·      Amoxicillin-clavulanic acid, cephalexin, and co-trimoxazole zone diameter breakpoints are not mentioned in the 2021 CLSI for testing Staphylococcus spp. susceptibility. How were the results interpreted? 

Please find below some minor comments

Abstract

Line 19: “pvl” should be changed to “pvl

Materials and methods 

Line 82: subheading should be changed to 2.2. Sample collection and phenotypic detection of S. aureus.

Line 83 “S. aureus” should be changed to “S. aureus”

Line 79: (Nasal, and ear, samples) should be changed to nasal, and ear swab samples.

Line 84 and 85: The protocols for coagulase and DNase biochemical tests should be briefly mentioned and references should be cited if present.

Lines 89-115: methods should be rearranged and restructured as follows:

2.3. DNA extraction and molecular detection of mecA and lukS/F-PV genes by PCR.                                                                                     

“DNA was extracted from cultured isolates using alkaline lysis, as previously described [26]. In brief, one bacterial colony was suspended in 20μl of lysis buffer (0.25% sodium dodecyl sulfate, 0.05 N NaOH) and heated for 15 minutes at 95 °C. 180 μl of distilled water was used to dilute the cell lysate. The cell debris was centrifuged at 16.000 x g for 5 minutes, and the supernatants were used for PCR or stored at -20 °C until further use. Extracted DNA from the different isolates was used in a multiplex PCR assay that detects three genes: the 16SrRNA gene used as an internal control, the mecA gene for MRSA detection, and lukS/F-PV gene for PVL detection [27]. 

Thermalcycling conditions set at 94°C for 10 min followed by 10 cycles of 94°C for 45 s, 111 55°C for 45 s, and 72°C for 75 s and 25 cycles of 94°C for 45 s, 50°C for 45 s, and 72°C for 112 75 s. PCR products were analyzed on 1.5% agarose gel containing 2μl ethidium bromide 113 (0.5 μg/mL). Bands with a size of 756-, 433-, and 310-bp correspond to 16S rRNA, lukS/F-114 PV, and mecA genes, respectively. The primers are listed in Table 1 [27].”

2.4. Antimicrobial susceptibility testing

Antibiotic susceptibility testing was performed by Kirby Bauer disc diffusion method as recommended by CLSI guidelines [25] using Mueller-Hinton agar. Prior to inoculation, the swab stick was dipped into bacterial suspension having visually equivalent turbidity to 0.5 McFarland standards. Antibiotic discs (Himedia, India) tested included penicillin G (10 μg), vancomycin (10 μg), erythromycin (15 μg), chloramphenicol (30 μg), co-tri-moxazole (25 μg), rifampicin (30 μg), clindamycin (2μg), amoxicillin-clavulanic acid  (20/10 μg), tetracycline (30 μg), ciprofloxacin (5 μg), cephalexin (30 μg) and cefoxitin (30 μg). Zone of inhibition for each antimicrobial agent was measured, and interpreted, as 

resistant, intermediate, or susceptible according to the CLSI guidelines [25]. Methicillin resistance was inferred using the zone diameter around cefoxitin following CLSI guidelines [25].

Line 90: “by” should be changed to “following”e Diameter Breakpoints

Line 97: “Zone of inhibition” should be changed to “The zones of inhibition”

Line 114-115: Sentence “Bands with a size of 756-, 433-, and 310-bp correspond to 16S rRNA, lukS/F-114 PV, and mecA genes, respectively” should be removed, and PCR products sizes should be listed with the primer sequences in Table 1

Line 116: The table title should be revised. 

Results 

Line 142: The table title should be revised. 

Line 143: The table title should be revised.

Line 145-148: rearrange the sentence and mention the antibiotics isolates are least susceptible to first.

Line 148: add erythromycin and its resistance rate to the antibiogram you are describing. 

Line 156: mention the criteria used for categorizing MDR and non-MDR S. aureus isolates.

Note that all MRSA isolates should be reported as MDR as recommended by the European Centre for Disease Prevention and Control (ECDC) and the Centers for Disease Control and Prevention (CDC). 10.1111/j.1469-0691.2011.03570.x

Discussion:

Line 160-167:  newer and more updated references should be cited instead of references [14] and [17].  The role PVL toxin plays in determining MRSA infection severity is controversial at best. Newer pieces of evidence show that the presence of pvl genes is neither the primary cause nor the determinant of MRSA infection severity.

https://doi.org/10.1038/s41579-018-0147-4

https://doi.org/10.1128/JCM.06421-11

https://doi.org/10.1371/journal.pone.0037212

Similarly, Reference [29] cited by the authors also states that the presence of pvl genes is not the main determinant of MRSA infection severity nor invasivity, 

Line 175: should mention that the higher prevalence of MRSA isolates in reference [33] is due to the detection method (oxacillin discs) chosen by the authors which is known to be less sensitive than molecular testing.

Line 177: “S. aureus” should be changed to “S. aureus”

Line 198: Should expand on the reasoning for the differences in PVL prevalence, one of which is due to the different dominant MRSA clones circulating these regions that could be or could be not harboring pvl genetic determinants.

Line 204: “Positivity” should be changed to “carriage”

Line 210-213: should add references.

Line 214-216: the high recovery rate should be attributed to the fact that most S. aureus isolates (208/285) were collected from pus clinical specimens. 

Lines 231-233: “study in Palestine that reported among MSSA isolates nearly complete resistance to penicillin G(98.9%) and to those of Bazzi et al. in Dahran – Saudi Arabia who reported 88.75% [55]” should be rewritten to become more concise 

Line 245-247: what is written contradicts what’s written in lines 155-156

Author Response

Reviewers 1

Query : The main focal point of this paper is about the association between PVL toxin and the severity of MRSA infections, even though this claim is highly controversial. While some studies have suggested that MRSA strains that produce PVL are more virulent and cause more severe infections, others have found no significant difference in the severity of infections caused by PVL-positive and PVL-negative strains. Furthermore, there is also evidence that PVL alone may not be the main factor associated with the severity of the infection and other virulence factors may play a more important role. Therefore, the relationship between PVL and the severity of MRSA infections is still a subject of ongoing research and debate and is not absolute. Therefore, the Title of this article should be redirected from PVL association and should focus more on the high prevalence of MRSA among S. aureus clinical isolates and the accompanying antibiotic resistance profiles identified. 

Answer : We agree that the role of PVL in pathogenicity is still debated, and we do not claim this. We analyzed the prevalence of mecA and pvl genes among S. aureus. For sake of consistency, we have added mecA in the title as well. The new title reads now « Prevalence of mecA and Panton-Valentine Leukocidingenes in Staphylococcus aureus clinical isolates from Gaza Strip hospitals »

Major comments/requests/questions

  • Query : The lack of molecular typing data about the isolates is a major limitation of this study. Therefore, the authors are requested to carry out sequence typing, spatyping and SCCmectyping for the isolates. This will provide a snapshot of the S. aureus strains circulating in Gaza and whether these strains are evolving. Also, the typing will provide some information about the origin of these isolates and whether they are likely to be community or hospital acquired. 

Answer : We agree with this comment, this is a clear limitation. However, these analyses (Sanger or Whole sequencing) are currently not possible in the Gaza Strip, and would require shipment of the isolates outside of the Gaza strip, which is not possible, or very complicated in a timely manner (several months of bureaucracy).

  • Query : According to the 2021 CLSI guideline, vancomycin antibiotic has no zone diameter breakpoints for differentiating Resistant isolates from susceptible ones. How were the results interpreted?

  Answer: We referred to 2021 CLSI guidelines when we followed the procedure of

  Antibiotic susceptibility testing using Bauer-Kirby Method but the interpretation of our results as sensitive/Intermediate /resistant was according to HIMEDIA technical data (which still uses the old CLSI breakpoints (17 mm)). This point has been clearly indicated.

  • Query : Amoxicillin-clavulanic acid, cephalexin, and co-trimoxazole zone diameter breakpoints are not mentioned in the 2021 CLSI for testing Staphylococcusspp. susceptibility. How were the results interpreted? 

Answer: We referred to 2021 CLSI guidelines when we followed the procedure of Antibiotic susceptibility testing using Bauer-Kirby Method but the interpretation of our results as sensitive/Intermediate resistant was according to HIMEDIA technical data (the company from which we purchased the antibiotic discs

Please find below some minor comments

Abstract

Query : Line 19: “pvl” should be changed to “pvl

Answer : corrected as suggested

Materials and methods 

Query : Line 82: subheading should be changed to 2.2. Sample collection and phenotypic detection of S. aureus.

Answer : corrected as suggested

Query : Line 83 “S. aureus” should be changed to “S. aureus”

Answer : corrected as suggested

Query : Line 79: (Nasal, and ear, samples) should be changed to nasal, and ear swab samples.

Answer : corrected as suggested

Query : Line 84 and 85: The protocols for coagulase and DNase biochemical tests should be briefly mentioned and references should be cited if present.

Answer: To confirm fermentation of mannitol, growth of yellow colonies on Mannitol salt agar (HiMedia, Mumbai, India) surrounded by yellow zones after 24 hours of incubation at 37°C indicated a positive result. For tube coagulase tests, colonies of test isolates were re-suspended in 2 ml of citrated human plasma in sterile glass test-tubes. The tubes were incubated at 35°C for 4 hours and observed for clot formation. DNase test was performed by inoculating Staph isolates on deoxyribonuclease (DNAse) agar (HiMedia, Mumbai, India) and incubated for 24 hours at 37°C. An excess of 1 N HCl (about 15 ml) was then added. A vacuum pipette was used to remove extra acid, and clear areas around the bacterial colonies revealed DNase-positive colonies (Kateete et al. 2010). This has been added lines 94-102

Query : Lines 89-115: methods should be rearranged and restructured as follows:

2.3. DNA extraction and molecular detection of mecA and lukS/F-PV genes by PCR.                                                                                     

“DNA was extracted from cultured isolates using alkaline lysis, as previously described [26]. In brief, one bacterial colony was suspended in 20μl of lysis buffer (0.25% sodium dodecyl sulfate, 0.05 N NaOH) and heated for 15 minutes at 95 °C. 180 μl of distilled water was used to dilute the cell lysate. The cell debris was centrifuged at 16.000 x g for 5 minutes, and the supernatants were used for PCR or stored at -20 °C until further use. Extracted DNA from the different isolates was used in a multiplex PCR assay that detects three genes: the 16SrRNA gene used as an internal control, the mecA gene for MRSA detection, and lukS/F-PV gene for PVL detection [27]. 

Thermalcycling conditions set at 94°C for 10 min followed by 10 cycles of 94°C for 45 s, 111 55°C for 45 s, and 72°C for 75 s and 25 cycles of 94°C for 45 s, 50°C for 45 s, and 72°C for 112 75 s. PCR products were analyzed on 1.5% agarose gel containing 2μl ethidium bromide 113 (0.5 μg/mL). Bands with a size of 756-, 433-, and 310-bp correspond to 16S rRNA, lukS/F-114 PV, and mecA genes, respectively. The primers are listed in Table 1 [27].”

Answer : corrected as suggested

2.4. Antimicrobial susceptibility testing

Antibiotic susceptibility testing was performed by Kirby Bauer disc diffusion method as recommended by CLSI guidelines [25] using Mueller-Hinton agar. Prior to inoculation, the swab stick was dipped into bacterial suspension having visually equivalent turbidity to 0.5 McFarland standards. Antibiotic discs (Himedia, India) tested included penicillin G (10 μg), vancomycin (10 μg), erythromycin (15 μg), chloramphenicol (30 μg), co-tri-moxazole (25 μg), rifampicin (30 μg), clindamycin (2μg), amoxicillin-clavulanic acid  (20/10 μg), tetracycline (30 μg), ciprofloxacin (5 μg), cephalexin (30 μg) and cefoxitin (30 μg). Zone of inhibition for each antimicrobial agent was measured, and interpreted, as 

resistant, intermediate, or susceptible according to the CLSI guidelines [25]. Methicillin resistance was inferred using the zone diameter around cefoxitin following CLSI guidelines [25].

Answer : corrected as suggested

Query : Line 90: “by” should be changed to “following”e Diameter Breakpoints

Answer : corrected as suggested

Query : Line 97: “Zone of inhibition” should be changed to “The zones of inhibition”

Answer : corrected as suggested

Query : Line 114-115: Sentence “Bands with a size of 756-, 433-, and 310-bp correspond to 16S rRNA, lukS/F-114 PV, and mecA genes, respectively” should be removed, and PCR products sizes should be listed with the primer sequences in Table 1

Answer : Deleted from text and added to table1

Query : Line 116: The table title should be revised. 

Answer : The title has been changed to ‘PCR primers used in this study’

Results 

Query : Line 142: The table title should be revised. 

Answer : The table has been changed to ‘Prevalence of mecA and pvl genes according to the hospitals in Gaza)

Query : Line 143: The table title should be revised.

Answer : The table has been changed to ‘Prevalence of mecA and pvl genes according to the sample type’

Query : Line 145-148: rearrange the sentence and mention the antibiotics isolates are least susceptible to first.

Answer : Done accordingly

Query : Line 148: add erythromycin and its resistance rate to the antibiogram you are describing. 

Answer : the resistance rate of erythromycin has been added to the text, as suggested.

Query : Line 156: mention the criteria used for categorizing MDR and non-MDR S. aureus isolates.

Note that all MRSA isolates should be reported as MDR as recommended by the European Centre for Disease Prevention and Control (ECDC) and the Centers for Disease Control and Prevention (CDC). 10.1111/j.1469-0691.2011.03570.x

 Answer : We agree that according to CDC, all MRSA should be reported as MDR isolates. However, here we have used the following definition to infer MDR, which resistance to at least three families of antibiotics. This point has been clearly stated in the text (line 170)

Discussion:

Query : Line 160-167:  newer and more updated references should be cited instead of references [14] and [17].  The role PVL toxin plays in determining MRSA infection severity is controversial at best. Newer pieces of evidence show that the presence of pvl genes is neither the primary cause nor the determinant of MRSA infection severity.

https://doi.org/10.1038/s41579-018-0147-4

https://doi.org/10.1128/JCM.06421-11

https://doi.org/10.1371/journal.pone.0037212

Answer : Thank you very much for pointing out this points. We tempered our discussion, and clearly cited the controversial situation (see line 180-191 ).

Query : Similarly, Reference [29] cited by the authors also states that the presence of pvl genes is not the main determinant of MRSA infection severity nor invasivity, 

Answer : This point has also been clarified (line 187-189)

Query : Line 175: should mention that the higher prevalence of MRSA isolates in reference [33] is due to the detection method (oxacillin discs) chosen by the authors which is known to be less sensitive than molecular testing.

Answer : This point has been added to the discussion as suggested (lines 199-201)

Query : Line 177: “S. aureus” should be changed to “S. aureus”

Answer : Done accordingly

Query : Line 198: Should expand on the reasoning for the differences in PVL prevalence, one of which is due to the different dominant MRSA clones circulating these regions that could be or could be not harboring pvl genetic determinants.

Answer : This point has also been added to the discussion (line 217-219)

Query : Line 204: “Positivity” should be changed to “carriage”

Answer : Done accordingly (line 232 and 250)

Query : Line 210-213: should add references.

Answer : Done accordingly. Refence 51 was added.

Query : Line 214-216: the high recovery rate should be attributed to the fact that most S. aureus isolates (208/285) were collected from pus clinical specimens. 

Answer : We agree and this has been clearly stated in the MS ‘The majority of our PVL positive isolates were isolated from pus and blood samples (72% and 64% respectively).’

Query : Lines 231-233: “study in Palestine that reported among MSSA isolates nearly complete resistance to penicillin G(98.9%) and to those of Bazzi et al. in Dahran – Saudi Arabia who reported 88.75% [55]” should be rewritten to become more concise Answer : This sentence has been rewritten as suggested ans reads now: “These results are comparable to those of Al Laham et al. [20] in Palestine and Bazzi et al. in Dahran – Saudi Arabia, that reported 98.9% and 88.75% resistance , respectively [55]. »

Query : Line 245-247: what is written contradicts what’s written in lines 155-156

Answer : Line 155-156, global susceptibility to vancomycin is indicated, whereas lines 245-247, susceptibility to vancomycin of MRSA is indicated.

Reviewer 2 Report

Thanks for allowing me to review your manuscript on the "Prevalence of Panton-Valentine Leukocidin in Staphylococcus  aureus clinical isolates from Gaza Strip hospitals"

Overall, I found this a very well written study. It is obviously important to assess the levels of antibiotic resistance especially to pathogens such as S. aureus. The only thing I felt that was missing in the discussion of the results was a reason or speculation as to why levels of resistant pathogens are so high. This could be an interesting start to future research. Or perhaps to find out which areas could be related to the best therapeutic outcomes.

Other than this, congratulations on a well written article.

There are a couple of typo's I noticed and a few comments below.

Line 18. Should be 'an important virulence factor'.

Line 20. Are the authors going to use Palestinia or Palestine ?

Line 32. What kind of interventions would the authors recommend ?

Line 58. Why Skin with a capital S

Line 81. Is there a study approval number ?

Author Response

Reviewer #2

Thanks for allowing me to review your manuscript on the "Prevalence of Panton-Valentine Leukocidin in Staphylococcus  aureus clinical isolates from Gaza Strip hospitals"

Overall, I found this a very well written study. It is obviously important to assess the levels of antibiotic resistance especially to pathogens such as S. aureus. The only thing I felt that was missing in the discussion of the results was a reason or speculation as to why levels of resistant pathogens are so high. This could be an interesting start to future research. Or perhaps to find out which areas could be related to the best therapeutic outcomes. 

Other than this, congratulations on a well written article. 

Answer : We are very thankful to this reviewer for these encouraging words.

There are a couple of typo's I noticed and a few comments below.

Query : Line 18. Should be 'an important virulence factor'.

Answer : corrected as suggested.

Query : Line 20. Are the authors going to use Palestinia or Palestine ?

Answer : corrected as suggested.

Query : Line 32. What kind of interventions would the authors recommend ?

Answer : With interventions we meant simple things, but yet not necessarily easy to do in Gaza strip hospitals, such as increased hand hygiene, use of hydroalcoholic solutions, and isolation of carriers.

Query : Line 58. Why Skin with a capital S

Answer : This is an error, and corrected accordingly

Query : Line 81. Is there a study approval number ?

Answer : Yes we have the approval number of the study PHRC/HC/227/17. This has been added to the text.

Reviewer 3 Report

Dear Authors,

I have read carefully the manuscript "Prevalence of mecA and Panton-Valentine Leukocidin genes in Staphylococcus aureus clinical isolates from Gaza Strip hospitals." The analysis of S. aureus isolates carrying the pvl genes and the methicillin resistance gene isolated from human samples is an interesting study. Although the subject of research is not very innovative, as the available literature contains numerous studies on the characteristics of S. aureus in humans, especially from hospitalized individuals. However, this particular study concerns strains of bacteria isolated in a specific area - the Gaza Strip. Since the tested strains were isolated from human tissues and secretions, the obtained results provide knowledge about the level of antibiotic resistance and some virulence factors of S. aureus causing infections in humans. In the context of the presence of methicillin resistance genes, I feel unsatisfied that the authors examined only the presence of the mecA gene in bacterial isolates and did not attempt to analyze the presence of the mecC gene. This homologue of the mecA gene, designated mecC, poses diagnostic problems with the potential to be misdiagnosed as methicillin-sensitive S. aureus, with important potential consequences for individual patients and for the surveillance of MRSA. The emergence of mecC MRSA is currently a topic of interest to human and veterinary microbiology. especially in the context of the occurrence of mecC MRSA strains in many not only European countries with different host species. The authors examined 285 strains of S. aureus from five different hospitals in Gaza Strip, which appears to be a sufficiently representative sample. The workflow is correct. The executive summary is well written and contains relevant information. Nevertheless, the manuscript should be corrected in some points to improve its quality.

In abstract: Lines: 30-31, the sentence: "The least effective antibiotics were penicillin and amoxicillin-clavulanic acid with resistance rates of 96.1% and 73.6%, respectively" should be rephrased as: The highest percentage of strains was observed to be resistant to penicillin and amoxicillin with clavulanic acid, 96.1% and 73.6%, respectively.

In the Introduction, the authors describing the mecA resistance gene but should also mention a divergent mecA homologue named mecC (formerly mecALGA251).

The material and methods were well planned. It is noteworthy that the authors chose cefoxitin and not oxacillin to assess the susceptibility of strains to antibiotics by the disc diffusion method, especially in the context of the lack of identification of the homologue -mecC. As some researchers point out, the majority of mecC MRSA show resistance to cefoxitin, and are therefore reported as MRSA, but however show susceptibility to oxacillin.

Nothing is mentioned about the positive control strains in the study leukocidin Panton-Valentine (lukF/S-PV), and mecA. The absence of positive controls (hence possible inappropriate PCR reaction) may explain the high detection rate.

Alternatively, multi-resistant strains can be genotyped. Without this, it is not known whether these strains are of clonal origin or not. MLST can be used to determine whether MR isolates are community acquired CA-MRSA or hospital acquired HA-MRSA.

In conclusion, I consider the manuscript to be well edited and, with minor corrections, ready for publication.

I have read carefully the manuscript "Prevalence of mecA and Panton-Valentine Leukocidin genes in Staphylococcus aureus clinical isolates from Gaza Strip hospitals." The analysis of S. aureus isolates carrying the pvl genes and the methicillin resistance gene isolated from human samples is an interesting study. Although the subject of research is not very innovative, as the available literature contains numerous studies on the characteristics of S. aureus in humans, especially from hospitalized individuals. However, this particular study concerns strains of bacteria isolated in a specific area - the Gaza Strip. Since the tested strains were isolated from human tissues and secretions, the obtained results provide knowledge about the level of antibiotic resistance and some virulence factors of S. aureus causing infections in humans. In the context of the presence of methicillin resistance genes, I feel unsatisfied that the authors examined only the presence of the mecA gene in bacterial isolates and did not attempt to analyze the presence of the mecC gene. This homologue of the mecA gene, designated mecC, poses diagnostic problems with the potential to be misdiagnosed as methicillin-sensitive S. aureus, with important potential consequences for individual patients and for the surveillance of MRSA. The emergence of mecC MRSA is currently a topic of interest to human and veterinary microbiology. especially in the context of the occurrence of mecC MRSA strains in many not only European countries with different host species. The authors examined 285 strains of S. aureus from five different hospitals in Gaza Strip, which appears to be a sufficiently representative sample. The workflow is correct. The executive summary is well written and contains relevant information. Nevertheless, the manuscript should be corrected in some points to improve its quality.

In abstract: Lines: 30-31, the sentence: "The least effective antibiotics were penicillin and amoxicillin-clavulanic acid with resistance rates of 96.1% and 73.6%, respectively" should be rephrased as: The highest percentage of strains was observed to be resistant to penicillin and amoxicillin with clavulanic acid, 96.1% and 73.6%, respectively.

In the Introduction, the authors describing the mecA resistance gene but should also mention a divergent mecA homologue named mecC (formerly mecALGA251).

The material and methods were well planned. It is noteworthy that the authors chose cefoxitin and not oxacillin to assess the susceptibility of strains to antibiotics by the disc diffusion method, especially in the context of the lack of identification of the homologue -mecC. As some researchers point out, the majority of mecC MRSA show resistance to cefoxitin, and are therefore reported as MRSA, but however show susceptibility to oxacillin.

Nothing is mentioned about the positive control strains in the study leukocidin Panton-Valentine (lukF/S-PV), and mecA. The absence of positive controls (hence possible inappropriate PCR reaction) may explain the high detection rate.

Alternatively, multi-resistant strains can be genotyped. Without this, it is not known whether these strains are of clonal origin or not. MLST can be used to determine whether MR isolates are community acquired CA-MRSA or hospital acquired HA-MRSA.

In conclusion, I consider the manuscript to be well edited and, with minor corrections, ready for publication.

Author Response

Query : I have read carefully the manuscript "Prevalence of mecA and Panton-Valentine Leukocidin genes in Staphylococcus aureus clinical isolates from Gaza Strip hospitals." The analysis of S. aureus isolates carrying the pvl genes and the methicillin resistance gene isolated from human samples is an interesting study. Although the subject of research is not very innovative, as the available literature contains numerous studies on the characteristics of S. aureus in humans, especially from hospitalized individuals. However, this particular study concerns strains of bacteria isolated in a specific area - the Gaza Strip. Since the tested strains were isolated from human tissues and secretions, the obtained results provide knowledge about the level of antibiotic resistance and some virulence factors of S. aureus causing infections in humans. In the context of the presence of methicillin resistance genes, I feel unsatisfied that the authors examined only the presence of the mecA gene in bacterial isolates and did not attempt to analyze the presence of the mecC gene. This homologue of the mecA gene, designated mecC, poses diagnostic problems with the potential to be misdiagnosed as methicillin-sensitive S. aureus, with important potential consequences for individual patients and for the surveillance of MRSA. The emergence of mecC MRSA is currently a topic of interest to human and veterinary microbiology. especially in the context of the occurrence of mecC MRSA strains in many not only European countries with different host species. The authors examined 285 strains of S. aureus from five different hospitals in Gaza Strip, which appears to be a sufficiently representative sample. The workflow is correct. The executive summary is well written and contains relevant information. Nevertheless, the manuscript should be corrected in some points to improve its quality.

Answer : We agree, that seeking for mecC would have been an important added value. In future studies, this resistance determinant will be included.

Query : In abstract: Lines: 30-31, the sentence: "The least effective antibiotics were penicillin and amoxicillin-clavulanic acid with resistance rates of 96.1% and 73.6%, respectively" should be rephrased as: The highest percentage of strains was observed to be resistant to penicillin and amoxicillin with clavulanic acid, 96.1% and 73.6%, respectively.

Answer : The sentence has been rephrased accordingly

Query : In the Introduction, the authors describing the mecA resistance gene but should also mention a divergent mecA homologue named mecC (formerly mecALGA251).

Answer : In the introduction, a new paragraph about mecC has been added

Query : The material and methods were well planned. It is noteworthy that the authors chose cefoxitin and not oxacillin to assess the susceptibility of strains to antibiotics by the disc diffusion method, especially in the context of the lack of identification of the homologue -mecC. As some researchers point out, the majority of mecC MRSA show resistance to cefoxitin, and are therefore reported as MRSA, but however show susceptibility to oxacillin.

Answer : These remarks will be taken into consideration in future studies about MRSA in Gaza Strip

Query : Nothing is mentioned about the positive control strains in the study leukocidin Panton-Valentine (lukF/S-PV), and mecA. The absence of positive controls (hence possible inappropriate PCR reaction) may explain the high detection rate.

Answer : We used positive control strains in the study. S. aureus ATCC 25923 (carrying the pvl gene), and S. aureus ATCC 33592 (carrying the mecA gene) were added in each run. This has been added to materials and methods section.

Query : Alternatively, multi-resistant strains can be genotyped. Without this, it is not known whether these strains are of clonal origin or not. MLST can be used to determine whether MR isolates are community-acquired CA-MRSA or hospital-acquired HA-MRSA.

Answer: Yes, we totally agree with your comment, but this technique is not available in Gaza. Hope in the future we can have such a facility in our Labs.

Query : I have read carefully the manuscript "Prevalence of mecA and Panton-Valentine Leukocidin genes in Staphylococcus aureus clinical isolates from Gaza Strip hospitals." The analysis of S. aureus isolates carrying the pvl genes and the methicillin resistance gene isolated from human samples is an interesting study. Although the subject of research is not very innovative, as the available literature contains numerous studies on the characteristics of S. aureus in humans, especially from hospitalized individuals. However, this particular study concerns strains of bacteria isolated in a specific area - the Gaza Strip. Since the tested strains were isolated from human tissues and secretions, the obtained results provide knowledge about the level of antibiotic resistance and some virulence factors of S. aureus causing infections in humans. In the context of the presence of methicillin resistance genes, I feel unsatisfied that the authors examined only the presence of the mecA gene in bacterial isolates and did not attempt to analyze the presence of the mecC gene. This homologue of the mecA gene, designated mecC, poses diagnostic problems with the potential to be misdiagnosed as methicillin-sensitive S. aureus, with important potential consequences for individual patients and for the surveillance of MRSA. The emergence of mecC MRSA is currently a topic of interest to human and veterinary microbiology. especially in the context of the occurrence of mecC MRSA strains in many not only European countries with different host species. The authors examined 285 strains of S. aureus from five different hospitals in Gaza Strip, which appears to be a sufficiently representative sample. The workflow is correct. The executive summary is well written and contains relevant information. Nevertheless, the manuscript should be corrected in some points to improve its quality.

Answer : We agree, that seeking for mecC would have been an important added value. In future studies, this resistance determinant will be included.

Query : In abstract: Lines: 30-31, the sentence: "The least effective antibiotics were penicillin and amoxicillin-clavulanic acid with resistance rates of 96.1% and 73.6%, respectively" should be rephrased as: The highest percentage of strains was observed to be resistant to penicillin and amoxicillin with clavulanic acid, 96.1% and 73.6%, respectively.

Answer : The sentence has been rephrased accordingly

Query : In the Introduction, the authors describing the mecA resistance gene but should also mention a divergent mecA homologue named mecC (formerly mecALGA251).

Answer : In the introduction, a new paragraph about mecC has been added

Query : The material and methods were well planned. It is noteworthy that the authors chose cefoxitin and not oxacillin to assess the susceptibility of strains to antibiotics by the disc diffusion method, especially in the context of the lack of identification of the homologue -mecC. As some researchers point out, the majority of mecC MRSA show resistance to cefoxitin, and are therefore reported as MRSA, but however show susceptibility to oxacillin.

Answer : These remarks will be taken into consideration in future studies about MRSA in Gaza Strip

Query : Nothing is mentioned about the positive control strains in the study leukocidin Panton-Valentine (lukF/S-PV), and mecA. The absence of positive controls (hence possible inappropriate PCR reaction) may explain the high detection rate.

Answer : We used positive control strains in the study. S. aureus ATCC 25923 (carrying the pvl gene), and S. aureus ATCC 33592 (carrying the mecA gene) were added in each run. This has been added to materials and methods section.

Query : Alternatively, multi-resistant strains can be genotyped. Without this, it is not known whether these strains are of clonal origin or not. MLST can be used to determine whether MR isolates are community-acquired CA-MRSA or hospital-acquired HA-MRSA.

Answer: Yes, we totally agree with your comment, but this technique is not available in Gaza. Hope in the future we can have such a facility in our Labs.

Query : In conclusion, I consider the manuscript to be well-edited and, with minor corrections, ready for publication.

Answer: Thank you very much. We thank you also for your valuable reviewing comments.

Query : In conclusion, I consider the manuscript to be well-edited and, with minor corrections, ready for publication.

Answer: Thank you very much. We thank you also for your valuable reviewing comments.

Reviewer 4 Report

The publication concerns an extremely important aspect of monitoring the spread of antibiotic resistance in pathogenic bacteria and detecting factors that increase their virulence. The authors tested S. aureus isolated from clinical samples for phenotypic antibiotic resistance, the presence of the mecA gene and the pvl virulence factor gene. The authors clearly presented the objectives of the work, described the methods used and presented the results of the research. In the discussion, the authors exhaustively touched on all aspects related to the scope of the described research.

Minor remarks

Line 110 and Table 1 “16S rRNA”

Table 4 – please provide an explanation for R, S and I.

Author Response

Minor remarks

Query : Line 110 and Table 1 “16S rRNA”

Answer : Has been corrected accordingly

Query : Table 4 – please provide an explanation for R, S and I.

Answer : Has been done Accordingly

Round 2

Reviewer 1 Report

Although this is an interesting study but not enough findings are presented to warrant publication as a full article. The lack of molecular typing data about the isolates is a major issue. Also, interpretation of the AST data should be done using the latest versions of the internationally recognized guidelines, such as the CLSI and EUCAST

Author Response

Query: Although this is an interesting study but not enough findings are presented to warrant publication as a full article. The lack of molecular typing data about the isolates is a major issue.

Answer: We agree that spa typing, and MLST should have been done, however, sequencing is not readily available in the Gaza strip, and sending isolates outside of the Gaza strip is very complicated, if not impossible given the political crisis this region of the world is going through. The study was conducted with what was available in the local research lab. Even though with shortcomings,  this study provides interesting results in terms of high prevalence of MRSA and PVL in the Gaza strip hospitals. Palestinian researchers and physicians are eager to fight against AMR in their country, even with very limited means.

Query: Also, interpretation of the AST data should be done using the latest versions of the internationally recognized guidelines, such as the CLSI and EUCAST

Answer: This is another important point, which is very complicated to address in the Gaza strip. Indeed, for some antibiotics, CLSI or Eucast advocate to do MIC testings, if these antibiotics are to be used in clinics. This is however not possible (or very complicated due to ressource limitations in the hospitals and research institutions). In Gaza strip hospitals, the use of antibiotics to treat bacteria infections is decided based on disk diffusion results. This point has been clearly stated in the text of the Manuscript. In fact, when breakpoints are available, CLSI guidelines are used ,and for those without breakpoints, either former breakpoints are still used or breakpoints provided by the manufacturer of antibiotic disk are considered. 

The following sentence has been added to the material and method section:

"The zones of inhibition for each antimicrobial agent were measured, and interpreted, as resistant, intermediate, or susceptible according to local interpretation rules used in the hospitals of the Gaza strip, which are derived from the CLSI guidelines when available,  from the technical data sheets of Himedia, or correspond to former breakpoints, which are R<10 mm and S>=16 mm for Bactrim; R<19 mm and S>=20 mm for Amoxicillin-clavulanic acid; R<12 mm and S>=18 mm for cefalexin; and R<17 mm and S>=17 mm for vancomycin"

Round 3

Reviewer 1 Report

Please remove the AST results that are not interpreted according the CLSI and EUCAST guidelines. 

I recommend using the following references to expand the citation list and compare the strains investigated in this study with other published study from the middle east

https://doi.org/10.3390/antibiotics12010078

10.2807/1560-7917.ES.2018.23.34.1700592

10.1002/nmi2.47

10.1111/jam.15526

10.1016/j.nmni.2016.03.008

10.1016/j.nmni.2016.07.009

Author Response

Query : Please remove the AST results that are not interpreted according the CLSI and EUCAST guidelines. 

Answer : It is clearly stated now in the manuscript that for those antibiotics, AST results were not interpreted according the CLSI nor EUCAST guidelines, but according to local criteria still in use in Gaza hospitals to treat patients. It is important to keep these values, for local epidemiological purposes. We have removed the values for cefalexin, and kept only Augmentin, vancomycin and Bactrim, as these are key molecules in Gaza hospitals. I agree, that these molecules may be not used in the best way, as susceptibility testing is not appropriate, but it is still better than no susceptibility at all.

Query : I recommend using the following references to expand the citation list and compare the strains investigated in this study with other published study from the middle east

https://doi.org/10.3390/antibiotics12010078

10.2807/1560-7917.ES.2018.23.34.1700592

10.1002/nmi2.47

10.1111/jam.15526

10.1016/j.nmni.2016.03.008

10.1016/j.nmni.2016.07.009

Answer : Thank you for your kind recommendation using these references. These published studies from the middle east mainly focused on molecular typing of clinical S. aureus using different methods and techniques that are not available in Gaza Strip and does not used in this study as DNA microarray, PFGE, SCCmec typing, Spa typing, MLST typing and WGS. However, most of them addressed the prevalence of the pvl gene. So, we -as per your recommendation- used some of them to expand the citation list and to compare their findings with our results. However, we already used and addressed many related studies in our discussion from the region and from around the world.

Please see lines 242-248 and the reference list in the updated version of our manuscript.